

# 1 Detection and origin of different types of annual laminae in
# 2 recent stalagmites from Zoolithencave, southern Germany:
# 3 Evaluation of the potential for quantitative reconstruction
# 4 of past precipitation variability

**Dana Felicitas Christine Riechelmann[1], Jens Fohlmeister[2], Rik Tjallingii[3,4],**
**Klaus Peter Jochum[5], Detlev Konrad Richter[6], Geert-Jan A. Brummer[4] and**
**Denis Scholz[1]**
[1]{Johannes Gutenberg-University Mainz, Institute for Geosciences, Johann-Joachim-
Becher-Weg 21, D-55128 Mainz, Germany}
[2]{Ruprecht-Karls University Heidelberg, Institute for Environmental Physics, Im
Neuenheimer Feld 229, D-69120 Heidelberg, Germany}
[3]{GFZ German Research Centre for Geosciences, Section Climate Dynamics and
Landscape Evolution, Telegrafenberg Building C, D-14473 Potsdam, Germany}
[4]{NIOZ-Royal Netherlands Institute for Sea Research, Department of Marine Geology and
Chemical Oceanography, Landsdiep 4, NL-1797 SZ 't Horntje (Texel), The Netherlands}
[5]{Max Planck Institute for Chemistry, Climate Geochemistry Department, Hahn-Meitner-
Weg 1, D-55128 Mainz, Germany}
[6]{Ruhr-University Bochum, Institute for Geology, Mineralogy and Geophysics,
Univeristätsstrasse 150, D-44801 Bochum, Germany}
Correspondence to: D. F. C. Riechelmann (riechelm@uni-mainz.de)

## 25 Abstract

An arrangement of three stalagmites from Zoolithencave (southern Germany) was analysed
for different types of annual laminae using both microscopic and geochemical methods. The
speleothems show visible laminae (consisting of a clear and a brownish, pigmented layer pair)



as well as fluorescent and elemental laminae. The age of the speleothems was constrained to
1800 to 1970 AD by $^{14}$C-dating of a charcoal piece below the speleothems, detection of the
$^{14}$C bomb peak, as well as counting of annual laminae. Dating by the $^{230}$Th/U-method was
impossible due to detrital contamination.
On the annual time-scale, the variability of Mg, Ba, and Sr is controlled by Prior Calcite
Precipitation (PCP) resulting in lower values during the wet season (autumn/winter) and vice
versa. Yttrium and P are proxies for soil activity and are enriched in the brownish, pigmented
layers. However, Y and P are also influenced by detrital content superimposing the soil
activity signal. Aluminium and Mn are proxies for detrital content.
Lamina thickness shows a significant correlation with the amount of precipitation of previous
December and current January, February, March, April, May, and December (DJFMAMD)
recorded at the nearby meteorological station Bamberg. Thus lamina thickness is a proxy for
past precipitation, which is confirmed by the good agreement with a precipitation
reconstruction based on tree-ring width from the Bavarian forest. This highlights the potential
of these speleothems for climate reconstruction at annual resolution.

## 1  Introduction

In the last decade, archives containing high-resolution climate proxies, which reflect past
climate variability on a time-scale relevant for civilisation, have achieved increasing attention
(e.g., Büntgen et al., 2011; Kennett et al., 2012). High-resolution climate reconstructions for
central Europe are mostly available from tree-rings span the last 2000 years (e.g., Wilson et
al., 2005; Büntgen et al., 2008; Trouet et al., 2009; Esper et al., 2012). Tree-ring records
covering longer time spans such as the entire Holocene are rare (Spurk et al., 2002; Friedrich
et al., 2004) and hold the problem of preserving low frequency climate signals. Speleothems,
such as stalagmites and flowstones, can grow continuously for several thousand years. They
can be precisely dated by the $^{230}$Th/U-method (e.g., Richards and Dorale, 2003; Scholz and
Hoffmann, 2008) and provide up to annual-resolution climate proxies. Therefore, they have
large potential to extend the existing tree-ring records (Tan et al., 2006). Some speleothems
show annual laminae and have the potential for annually or even seasonally resolved climate
reconstruction (Brook et al., 1999; Proctor et al., 2002; Boch and Spötl, 2008; Mattey et al.,
2008; Hardt et al., 2010; Orland et al., 2012; Myers et al., 2015; Ridley et al., 2015).





Typically, five types of annual laminae can be observed in speleothems: i) visible laminae
with a white and a dark/clear layer representing one year (Genty and Quinif, 1996; Scholz et
al., 2012; Van Rampelbergh et al., 2014), ii) fluorescent laminae induced by humic and fulvic
acid (van Beynen et al., 2001; Proctor et al., 2002; Shopov, 2003; Sundqvist et al., 2005), iii)
elemental laminae visible in cyclic (seasonal) changes in the concentration of specific
elements (Roberts et al., 1998; Huang et al., 2001; Treble et al., 2003; Johnson et al., 2006;
Borsato et al., 2007; Smith et al., 2009), iv) stable carbon and oxygen isotope laminae visible
in changes in the $\delta^{13}C$ and $\delta^{18}O$ values over an annual cycle (Mattey et al., 2008; Baker et al.,
2011; Boch et al., 2011; Van Rampelbergh et al., 2014; Myers et al., 2015; Ridley et al.,
2015), and v) mineralogical laminae consisting of calcite-aragonite pairs representing one
year (Railsback et al., 1994; Baker et al., 2008). All types of laminae have been analyse in
several studies and their potential for reconstruction of climate variability evaluated. Visible
annual laminae in speleothems can be induced by cave ventilation, which controls the super
saturation of the drip water with respect to calcite via modulation of the $pCO_2$ of in the cave
air. Cave ventilation is controlled by the temperature difference between outside atmospheric
and cave air and may result in a temperature signal in $\delta^{13}C$ and $\delta^{18}O$ speleothem records
(Boch et al., 2011). In addition, the annual cycle in the concentration of Mg, Sr, Ba, and/or P
were used both as temperature (Mattey et al., 2008) or precipitation proxies (Roberts et al.,
1998; Huang et al., 2001; Treble et al., 2003). The lamina thickness of both visible and
fluorescent laminae were used as proxies for past precipitation and water excess (Genty and
Quinif, 1996; Baker et al., 1999; Brook et al., 1999; Proctor et al., 2000; Boch and Spötl,
2008) or temperature (Frisia et al., 2003; Scholz et al., 2012).
In this study, we analysed an arrangement of three small stalagmites from Zoolithencave,
southern Germany, for their visible, fluorescent, and elemental laminae. The aims of this
study are i) to test the potential of different analytical methods to detect annual laminae in
speleothems, ii) analyse the origin of the different types of laminae, and iii) evaluate their
potential as climate proxies.

## 2 Cave setting

Zoolithencave (49°47' N, 11°17' E) is located in the Franconian Alb, south-eastern Germany,
and developed in the Upper Jurassic Franconian Dolomite (Fig. 1). The dolomitisation of this
massive spongal reef limestone already started in the Upper Jurassic (Meyer, 1972). The first



karstification of the dolomite occurred predominantly along fissures with NW-SE and NE-SW
orientation during the uplift of the Franconian Alb at the transition from the Jurassic to the
Cretaceous. The main phase of karstification took place in the Quaternary and coincided with
the further uplift of the Franconian Alb and erosion by rivers (Groiß, 1988).
The Zoolithencave is famous for its paleontological inventory, which was first described by
Esper (1774) and Rosenmüller (1794). Bones of several Pleistocene mammals were found,
and the cave is the first location where cave bear (*Ursus spelaeus*) bones were found. Further
archaeological excavations found charcoal and pottery from the Iron Age, and an ash layer in
a flowstone was dated to the late Mesolithicum (Rosendahl, 2005). This proves human
utilisation of the cave. Zoolithencave was intensively studied during the late 18[th] to early 20[th]
century due to its paleontological inventory. The second phase of scientific investigation
started in 1971 when further parts of the cave where discovered and paleontological analyses
were performed by, for instance Groiß (1979) and Diedrich (2014). At the same time,
speleological studies were conducted (e.g., Tietz, 1988; Wurth et al., 2000; Wurth, 2002;
Richter et al., 2014; Riechelmann et al., 2014).
The cave entrance is located 455 m above sea level on the north-east facing slope of the Hohle
Berg. The peak of the Hohle Berg is 469.9 m above sea level and forms a small karst plateau.
The average rock overburden of the cave is 15-20 m, which is covered by soil consisting of a
15 cm thick humic A-horizon and a > 30 cm thick loamy B-horizon (Wurth, 2002). The
vegetation above the cave mainly consists of deciduous forest (i.e., predominately beech).

## 113   3    Material and Methods

### 114   3.1 Stalagmite Zoo-rez

Stalagmite Zoo-rez is an arrangement of two stalagmites with a distance of 7 cm between
their tops, which grew in entrance hall of Zoolithencave. Zoo-rez-1 has a height of 3 cm,
whereas Zoo-rez-2 is 2.7 cm high. A third, 2.5 cm-high stalagmite (Zoo-rez-3) grew at close
distance. All three stalagmites were sampled in August 1999 (Fig. 2; Wurth (2002)). The base
of the arrangement of Zoo-rez-1 and -2 consists of cave loam, as well as sinter and charcoal
pieces, which are consolidated by calcite. The stalagmite was fed by an active drip when it
was sampled suggesting recent growth.



### 3.2 Dating methods


Two samples (ca. 300 mg) from the top and the base of Zoo-rez-1 (Fig. 3a), respectively,
were drilled for $^{230}$Th/U-dating using a hand held dental drill. The samples were dissolved in
7N HNO$_3$ and spiked with a mixed $^{229}$Th-$^{233}$U-$^{236}$U spike solution. The Th and U fractions
were separated by ion-exchange column chemistry (see Yang et al. (2015), for details) and
subsequently analysed using a Nu Plasma MC-ICP mass-spectrometer (*Nu Instruments Ltd.,*
*Wrexham*) at the Max Planck Institute for Chemistry, Mainz. For further methodological and
analytical details, the reader is referred to Obert et al. (accepted).
All activity ratios and $^{230}$Th/U-ages were calculated using the half-lives of Cheng et al.
(2000). To account for potential detrital contamination, corrected ages were calculated
assuming an upper continental crust $^{232}$Th/$^{238}$U weight ratio of 3.8 ± 1.9 (Wedepohl, 1995)
and secular equilibrium between $^{230}$Th, $^{234}$U, and $^{238}$U.
A piece of charcoal found at the base of Zoo-rez-2 (Fig. 3b) was analysed by accelerator mass
spectrometer (AMS) $^{14}$C-dating at the Curt-Engelhorn-Zentrum for Archeometry gGmbH,
Mannheim, Germany. The sample was prepared with the ABA-method (HCl/NaOH/HCl),
whereat the insoluble components were burned and the resulting CO$_2$ catalytically reduced to
graphite. The $^{14}$C content was determined with a MICADAS accelerator mass spectrometer
(Synal et al., 2007). In addition, three calcite samples from the top of stalagmite Zoo-rez-1
(Fig. 3a) were analysed for their $^{14}$C content in order to detect the atmospheric $^{14}$C bomb
peak. These samples were dissolved in vacuo and subsequently reduced to graphite at 575°C
under a H$_2$ atmosphere. The measurements were performed using the same setup as for the
charcoal sample.

### 3.3    Fluorescence and polarisation microscopy and determination of lamina thickness


Fluorescence microscopy was performed on thin sections of 30 μm thick using a *Leica*
*DM4500P* microscope, which is equipped with a *Canon Eos 60D* camera at the Institute for
Geology, Mineralogy and Geophysics at Ruhr-University Bochum. For UV-luminescence, a
BP360/40 excitation filter, a dichromatic mirror of 400 nm and an LP425 suppression filter
were used. Polarisation microscopy was performed with a *Leica DM750P* microscope. The
thin sections were further scanned with a *Colorview I* camera (*Olympus*) installed on an



*Olympus EX51* microscope, which is equipped with a *Märzhäuser LPT15* microscope stage
and a *Plan N20x* objective resulting in high resolution pictures. Determination of lamina
thickness on these pictures was performed using the software analySIS pro (*Olympus Soft*
*Imaging Solutions*). On Zoo-rez-1, three tracks were measured. On Zoo-rez-2 and -3 one track
each was measured. The microscopy tracks followed the LA-ICPMS tracks (Fig. 3, see also
section 3.5). Cross dating of the lamina thickness of the different tracks was performed using
the tree-ring analysis software tools TSAP-Win® (*RINNTECH, Heidelberg*) and COFECHA
(Holmes, 1983).

### 163 3.4 LA-ICPMS measurements for elemental concentrations

The elemental concentration of the three stalagmites was determined with an Element2 ICP
mass spectrometer (*ThermoScientific, Waltham, USA*) equipped with a high energy Nd:YAG
laser ablation system ($\lambda = 213$ nm) (*New Wave, Fremont, USA*). The reference material used
for calibration was NIST SRM 612, a synthetic glass with a high trace element content
(Jochum et al., 2011). The spot size of the laser beam was 110 µm, the puls repetition rate 10
Hz and the scan speed 10 µm/s. The elemental concentrations were normalised using Ca as an
internal standard. In total, 33 elements were measured: Na, Mg, Al, Si, P, Ca, Ti, Mn, Fe, Cu,
Zn, Rb, Sr, Y, Cd, Ba, La, Ce, Pr, Nd, Sm, Eu, Gd, Tb, Dy, Ho, Er, Tm, Yb, Lu, Pb, Th and
U. Seven (Mg, Al, P, Mn, Sr, Y, Ba) revealed reliable concentrations well above the detection
limit. As for the microscopy tracks, three tracks on stalagmite Zoo-rez-1, and one track each
on Zoo-rez-2 and -3 were measured (Fig. 3).

### 176 3.5 UV-Luminescence Scanning

UV-luminescence of the three speleothems was measured with an *Aavatech* core scanner at
the NIOZ (Texel, Netherlands). Images were acquired with a resolution of 70 µm/pixel using
a JAI CCD camera, equipped with a beam splitter separating the red, green and blue colour
channels (Grove et al., 2010). The speleothem samples were irradiated with a 365 nm UV-
LED lamp to initiate the luminescence. The CCD camera was equipped with a 435 nm cut off
filter to avoid recording of reflecting light of the initial light source. RGB colour information
was obtained from the images along selected transects, corresponding to the elemental and
lamina thickness transects, using the software Line Scan 2.0 of the Avaatech scanner.






## 3.6 Data analysis

Principal Component Analysis (PCA) and wavelet analysis of the elemental data series were
performed using the software *PAST (Hammer et al., 2001)*. For both analyses, the data were
normalised. In addition, Pearson correlation coefficients (r) were calculated between the
different elemental data series as well as between the proxy series and different climate
parameters. Wiggle matching of the different tracks (Mg content, UV-luminescence, and
lamina thickness) was conducted using the software *AnalySeries (Paillard et al., 1996)*.
Interpolation of the elemental and UV-luminescence data series as well as the calculation of
mean curves were performed with R (R Core (Team, 2015). Detrending of the lamina
thickness and mean annual Mg records with a 10 point FFT (Fast Fourier Transformation)
filter was performed using Origin®.

## 4   Results

## 4.1 Visible laminae

Visible laminae in Zoo-rez appears as layer pairs consisting of a clear layer and a layer with
brownish pigmentation (Fig. 4a). Counting these laminae along the three tracks of Zoo-rez-1,
results in 124, 161, and 135 laminae, respectively. In Zoo-rez-2, we counted 165 laminae, and
in Zoo-rez-3, 144 laminae. The mean laminae thickness varies from 129 (Zoo-rez-2) to 203
μm (Zoo-rez-1, track 1). The minimum lamina thickness varies from 25 (Zoo-rez-1, track 2)
to 56 μm (Zoo-rez-1, tracks 1 and 3), whereas the maximum lamina thickness ranges from
388 (Zoo-rez-2) to 917 μm (Zoo-rez-3). Further microscopic analysis of the thin sections of
the three stalagmites did not provide any evidence for growth stops. Therefore, continuous
growth is assumed. The crystal fabric of all stalagmites is columnar facicular optic and only
some patches of Zoo-rez-1 show a columnar radiaxial fabric (Richter et al., 2011; Frisia,

210  2015).


## 4.2 Chronology of stalagmite Zoo-rez

The corrected [230]Th/U-ages determined for Zoo-rez-1 are of 4670 ± 1000 years BP (this refers
to 2014 AD) for the sample taken at 0.3 cm distance from top (dft) and 340 +3314/-295 years





BP for the sample from 1.4 cm dft (Table 1). The very large age uncertainties result from the
large degree of detrital correction, which results in differences between corrected and
uncorrected ages of up to 5000 years (Table 1). This is a result of the low U and elevated
$^{232}$Th content of the two samples and is also evident from a very low ($^{230}$Th/$^{232}$Th) activity
ratio, which range from 0.9 to 2.6 (Table 1). For ($^{230}$Th/$^{232}$Th) ratios lower than 20, a
correction for detrital contamination is necessary (Schwarcz, 1989). However, in particular
for very young samples, such as stalagmite Zoo-rez-1, the conventionally applied bulk Earth
correction is not adequate and more elaborate methods are required (Ku and Liang, 1984;
Schwarcz and Latham, 1989; Bischoff and Fitzpatrick, 1991; Przybylowicz et al., 1991;
Kaufman, 1993; Ludwig, 2003; Pons-Branchu et al., 2014; Wenz et al., in review). Thus, the
two $^{230}$Th/U-ages determined for stalagmite Zoo-rez-1 cannot be considered reliable, which is
also obvious from the age inversion (i.e., the age close to the surface is older than the age at
the base of the stalagmite, Table 1). As a consequence, other methods to establish the
chronology of stalagmite Zoo-rez have to be used.
The $^{14}$C-age of the charcoal piece from the base of stalagmite Zoo-rez-2 is 165±21 years BP
(refers to 1950 AD), with a calibrated 1σ-range of 1671-1951 AD (Table 2). Calibration was
performed with INTCAL13 (Reimer et al., 2013) and SwissCal 1.0 (L. Wacker, ETH-Zürich).
Furthermore, the $^{14}$C-activity of three samples from the top of stalagmite Zoo-rez-1 was
determined in order to detect the atmospheric bomb peak (Hua et al., 2013). This atmospheric
bomb peak was induced by the above ground atomic bomb tests in 1945-1963 AD. The $^{14}$C
from this tests was circulated worldwide by the atmosphere and reached for example
stalagmites via rain and drip water. The $^{14}$C activity detected in a speleothem is always lower
than in the atmosphere, due to dissolution of the hostrock which contains carbon as well. The
sample from 0.8 mm dft shows the highest $^{14}$C-activity (Fig. 5). The subsequent decrease in
atmospheric $^{14}$C-activity has not been observed in the stalagmite suggesting that Zoo-rez did
not grow until 1999 AD (the year of sampling). The maximum of speleothem bomb spikes
appears to be near the atmospheric peak as long as the increase in radiocarbon is large. For
speleothems with a smaller increase in $^{14}$C the maximum in the speleothem is delayed. This is
explained by the age spectrum of SOM (Fohlmeister et al., 2011). Since the increase in $^{14}$C in
Zoo-rez is large (compare e.g., Noronha et al., 2015), the peak is near the maximum of the
atmospheric $^{14}$C values. Thus, we suggest that the highest $^{14}$C value corresponds to about
1967 AD and attributed a 5 years uncertainty. It follows that the stalagmite stopped growing
around 1970 AD ± 5 years by adjusting the $^{14}$C sampling site by lamina counting. The age of



the charcoal, which must be older than the stalagmite growing on top, is in good agreement
with the number of 124 to 165 layers counted. Therefore, stalagmite Zoo-rez most likely grew
during the last 150-200 years and shows visible annual laminae.

**4.3 Elemental laminae**
Seven elements (Mg, Al, P, Mn, Sr, Y, Ba) revealed reliable concentrations well above the
detection limit. The element records obtained from the five sampling tracks were compared
by calculation of correlations that reveal in two groups of elements. The first group reveals
positive correlation with each other contains of the elements Mg, Sr, and Ba, whereas the
second group shows positive correlations with each other contains of Al, P, Mn, and Y (Table
3). This is confirmed by the results of the Principle Component Analysis (PCA; von Storch
and Zwiers, 2002; Navarra and Simoncini, 2010), performed with the normalised elemental
concentrations, with Mg, Sr, and Ba grouping together, as well as Al, P, Mn, and Y. Only for
the PCA of Zoo-rez-2 Al and Mn as well as P and Y form two different groups (Fig. 6).
The time series of Mg, P, Sr, Y, and Ba show a cyclicity with higher and lower values (Figs.
7a and b). Aluminium and Mn do not show this pattern (Fig. 7c). However, both elements
show extreme concentrations (i.e., spikes) in some sections of the speleothems and very low
concentrations in other parts (Fig. 7c). The observed cyclicity, which probably results from
annual variations in elemental supply, is most pronounced for Mg (Fig. 7a). Phosphorus, Sr,
Y, and Ba show several spikes superimposed on the cyclicity, which is not the case for Mg
(Figs. 7a and b). Therefore, potential annual elemental lamination seems to be most
pronounced for Mg. In order to test whether the observed cyclicity is annual, wavelet analysis
has been performed for the five Mg time series (cf., Smith et al., 2009). The five wavelet plots
show a continuous cyclicity in the range of 64 to 256 µm over the whole length of all
measuring tracks (Fig. 8), which is in agreement with thickness of the visible laminae
(compare section 4.1). This strongly suggests that the observed cyclicity of Mg concentration
reflects an annual signal. Due to the observed positive correlation and grouping in the PCA
between Mg, Ba, and Sr (Table 3), it is likely that the variability of all three elements reflect
an annual signal.

**4.4 Luminescent laminae**



The pixel resolution of UV-luminescence scanning does not allow an annual signal (cf., Fig.
14) as the mean lamina thickness of the visible layers observed for the five different tracks of
Zoo-rez is in the range of two pixels (compare section 4.1). However, this method is not really
suitable to detect annual laminae in speleothem Zoo-rez, but rather for multi-annual scale
fluctuations.
UV-luminescence microscopy clearly shows a lamination, with the brownish layers exhibiting
a stronger luminescence than the clear layers under UV light (Fig. 4). Pronounced brownish
layers provide a stronger and more easily detectable luminescence than less pronounced
layers, which confirms the observations of the visible laminae. Therefore, fluorescence
microscopy does not provide additional information for stalagmite Zoo-rez.

## 290     5    Interpretation and Discussion

### 291     5.1 Chronology

Based on the five laminae thickness tracks, a chronology with an annual resolution was built
by visual cross-dating using the tree-ring software TSAP-Win®. Due to the geometry of
stalagmite Zoo-rez-1, the laminae get thinner and even disappear with increasing distance
from the growth axis (Fig. 3a). This is especially the case for those sections of Zoo-rez-1
showing no clear plateau (Fig. 3a) and is probably also the major reason for the different
number of laminae counted for the three different tracks on Zoo-rez-1. Track 2 is closest to
the growth axis and should therefore have less missing laminae than the other tracks. This is
confirmed by the observed number of laminae (161 for track 2 and 124 and 135 for track 1
and 3, respectively). Consequently, tracks 1 and 3 were cross-dated to track 2, and the
corresponding number of missing laminae was inserted into the chronology using TSAP-Win.
We assumed a lamina thickness of 10 μm for the missing laminae, which is the accuracy of
the thickness determination. In total, 19 missing laminae were inserted into track 1, nine
laminae into track 2 and 11 into track 3. Also in the master track 2 nine laminae were inserted,
which most likely is due to irregularities in the continuous growth of the laminae over the
stalagmite surface. Furthermore, the total amount of laminae after the cross-dating differ for
the different tracks, which is due to more or less clear laminae structure near to the base part
of the stalagmites. We note that this procedure is a standard technique in tree-ring research
(Fritts, 1976; Schweingruber, 1983; Speer, 2010). Subsequently, the two individual tracks on



Zoo-rez-2 and -3 were cross-dated to the mean curve of the three tracks on Zoo-rez-1. Into
track Zoo-rez-2, six laminae were inserted, and into track Zoo-rez-3, eleven laminae were
inserted. As in tree-ring cross-dating, each inserted laminae is present in at least one of the
tracks and, thus not missing in all tracks. Subsequently to visual cross-dating of the five
tracks, the chronology was checked using the tree-ring software COFECHA (Holmes, 1983).
This check calculates the series intercorrelation, which is the mean of the correlations of each
series with the mean of the remaining series. For our chronology, the series intercorrelation is
0.51. Furthermore, COFECHA calculates the correlation of the series with the mean of the
other series to detect potential dating errors by shifting by maximum 10 years in both
direction using segments of 50 years with an overlap of 25 years. For some segments, the
correlation was lower than for the shifted segment. However, these correlation coefficients
were not significantly higher and do therefore, not suggest an error during cross-dating
(Holmes, 1983; Speer, 2010). In summary, the cross-dating procedure results in a chronology
of 171 years, which represents the mean annual lamina thickness of the five series. Note that
neither additional missing laminae nor laminae not representing a full year can be excluded.
However, due to the five series and the cross-dating, which do not show distinct dating errors,
this chronology can be considered as relatively robust.
To assign an absolute age to the floating chronology, the [14]C-ages were used (see section 4.2)
and assuming annual laminae in the stalagmite, the age of the uppermost layer was set to 1970
AD. This assumes a fast percolation of the rain water into the cave, which is supported by the
low (10-12 m) rock overburden of the entrance hall of Zoolithencave. A short residence time
in the aquifer is a general prerequisite for the formation of annual laminae in speleothems
(Baker et al., 2008). Another factor, potentially producing annual laminae in speleothems is
strong cave ventilation resulting in a strong annual variability of cave $pCO_2$ (Huang et al.,
2001; Mattey et al., 2008; Boch et al., 2011). However, this can be excluded for
Zoolithencave because monitoring results show that cave $pCO_2$ is relatively low (530-1662
ppmV) in the entrance hall and does not vary by more than 1000 ppmV throughout the year
(Meyer, 2014). Hence, the observed annual lamination in stalagmite Zoo-rez is most likely
related to annual changes in drip water composition (Roberts et al., 1998; Huang et al., 2001;
Treble et al., 2003; Wassenburg et al., 2012).
The age of the lowermost lamina of stalagmite Zoo-rez is 1800 AD, which is in a good
agreement with the calibrated 1σ-range of the [14]C dating of the charcoal resulting in 1671-
1951 AD, whereas one possible calibration range spikes around 1800 AD. Due to the good



agreement of the number of counted laminae with the [14]C-dating results, the number of
missing and/or double-counted years should be very low and can be neglected. Nevertheless,
the absolute age of the chronology may vary by a few years (cf., Shen et al., 2013). The
resulting annually resolved chronology for Zoo-rez (Fig. 9) can be further used to assign a
chronology to the proxy signals.

**5.2 Wiggle matching and data interpolation**
Due to the potential annual nature of Mg cycles, wiggle matching between the individual Mg
series was performed using the software *AnalySeries*. As for the visible laminae (compare
section 5.1), track Zoo-rez-1.2 was chosen as the master series, which shows the largest
number of visible layers and is closest to the growth axis (Fig. 3a). All other Mg tracks were
wiggle matched on this master track (Fig. 10), which leads to an increase in the correlation
coefficients between the individual tracks to of r = 0.43 to r = 0.49. This is substantially
higher than correlation coefficients prior to wiggle matching, which range from r = 0.10 to r =
0.24. On average, the data points on track Zoo-rez-1.1 were shifted by 179 µm, by 502 µm on
track Zoo-rez-1.3, by 344 µm on Zoo-rez-2, and by 1253 µm on Zoo-rez-3. The relatively
large shift on Zoo-rez-3 is probably related to the largest distance of this track from the master
track (Zoo-rez-1.2). Subsequently to wiggle matching, a mean curve of all five Mg signals
was calculated. This mean Mg series was cut off at the end of the shortest series (i.e., Zoo-rez-
2). Lamina thickness was determined on thin sections, which were produced from the
opposite sides of the slices used for the elemental measurements. Therefore, it is not possible
to use the individual lamina thickness chronologies to construct an age model for the Mg
individual signals. Thus, the lamina thickness chronology was wiggle matched to the mean
Mg curve. The laminae consist of a clear layer and brownish-pigmented layer. The clear layer
corresponds to higher Mg concentration and the brownish pigmented to lower Mg
concentration. The end of the brownish pigmented layer represents approximately the end of
the flush in of humic particles. Therefore, this boundary for the laminae in the mean Mg
record was set in the middle of the increasing slope of the cycle in the Mg concentration (Fig.
11). The average shifting of the laminae boundaries was 44 µm (Fig. 11). Since both the
lamina thickness and the Mg curve are based on five individual tracks, we consider the
chronology of the resulting proxy time series as relatively robust. Since the resolution of the
Mg curve is much higher than that of the lamina thickness series, the mean Mg concentration



of the individual years was calculated. This results in an annually resolved Mg time series
(Fig. 12).
The G/B ratio series of the UV-luminescence scanning analyses from the three tracks on Zoo-
rez-1 were wiggle matched in the same way as the Mg curves and a mean curve was
calculated. Due to the lower resolution of 70 μm/pixel of the G/B ratios, Mn, and Y
concentration series need to be interpolated on the scale of the G/B ratios to compare these
series.

**5.3 Interpretation of the proxy signals in terms of past climate variability**

*5.3.1 Elemental laminae*

Magnesium, Ba, and Sr concentration are significantly correlated with each other (Table 3)
and also form a group in PCA (Fig. 6). The Mg, Ba, and Sr content of Zoo-rez is higher in
spring and summer (drier conditions, clear laminae) and lower in autumn and winter (wetter
conditions, brownish laminae). This is probably induced by prior calcite precipitation (Treble
et al., 2003; Smith et al., 2009) occurring in air filled pockets and cavities in the aquifer above
the cave. PCP increases the Mg, Sr, and Ba content of the drip water and, hence, in
speleothem calcite (Fairchild et al., 2006). The occurrence of air filled cavities and pockets in
the karst aquifer is most pronounced in the summer season, which results in increased PCP
(Wassenburg et al., 2012). Therefore, Mg, Ba, and Sr concentration is a proxy for recharge of
the karst aquifer and directly linked to precipitation. These three elements, thus, reflect the
annual cycle of infiltration with higher amounts of infiltration during autumn and winter and
lower infiltration in spring and summer. During spring and summer, evapotranspiration
reduces the amount of infiltrating rain water (Wackerbarth et al., 2010; Mischel et al., 2015).
Magnesium and Sr concentration were also determined for a small section of another
stalagmite from Zoolithencave by Wurth (2002). The results show a higher Mg and Sr content
of the clear layers and a lower content of the brownish layers. The brownish layers probably
result from a flush of organic material into the cave during autumn (Huang et al., 2001;
Sundqvist et al., 2005) when the recharge of the aquifer increases after the summer, which is
characterised by strong evapotranspiration (Mischel et al., 2015).
The second group identified by the PCA consists of Al, Mn, P, and Y. An exception is Zoo-
rez-2, where Al and Mn as well as P and Y form two different groups (Fig. 6d). Phosphorus
and Y are interpreted as proxies for vegetation density and soil activity (Treble et al., 2003;



Borsato et al., 2007; Wassenburg et al., 2012). Therefore, their concentration should be
elevated in the brownish layers reflecting increasing recharge. Yttrium and P are positively
correlated in all tracks (Table 3), supporting this interpretation. For track Zoo-rez-2, a
negative correlation between Mg and Y is observed (Table 3 and Fig. 13) (cf., Mattey et al.,
2008). For the others tracks, this relationship is only observed for some sections. This is
probably a result of detrital contamination, which is also visible in the positive correlations of
P and Y with Al and Mn (Wassenburg et al., 2012) as well as in the grouping of these
elements in the PCA (Fig. 6), because Al and Mn are proxies for detrital material. In Zoo-rez-
2, only a low correlation of P and Y with Al and Mn is observed (Fig. 6d), suggesting that
Zoo-rez-2 contains less detrital material. This is supported by the observed lowest amounts of
Al and Mn of all tracks. Therefore, P and Y cannot be considered as pure proxies for soil
activity/precipitation, but also for detrital input. However, the detrital input seams not to be
regular as the input of humic particles. This is determined by the results of the fluorescence
microscopy, where the brownish pigmented layers show a stronger UV-luminescence (Fig.
4b). In the case of detrital input during the flushing phase of the year the brownish-pigmented
layers should not show a strong luminescence, because the detrital material appears dark in
the UV-luminescence. Furthermore, Mn and Al do not show a seasonal cyclicity, but show
very high concentrations in the detrital rich sections at approximately 2 and 15 mm dft
confirming their association with detrital material.
Magnesium shows the strongest seasonal cyclicity and is interpret as a proxy for the seasonal
recharge cycle. Therefore, we averaged the Mg concentration for each year by wiggle
matching the visible annual laminae to the Mg (see section 5.2). The resulting annually
resolved Mg series from 1839 to 1970 AD shows a positive correlation of $r = 0.22$ ($p < 0.05$)
with the lamina thickness chronology (Fig. 12). This correlation is not only due to the same
long-term trend, but also the year to year variability especially in the 20[th] century show a
correlation of $r = 0.26$ ($p < 0.05$) after detrending with a 10 point FFT (Fast Fourier
Transformation) filter. Since lamina thickness is also interpreted as a proxy for precipitation
(see section 5.3.3), this positive correlation is surprising, in particular as the annual Mg cycle
shows a negative correlation to infiltration. However, this positive correlation may be
explained as follows: Higher rainfall may induce more active vegetation, which results in
higher soil $pCO_2$ (Harper et al., 2005; Wassenburg et al., 2012; Borsato et al., 2015). This
$CO_2$ is dissolved in the seeping water and may result in an increased dissolution of the
hostrock, which in the case of Zoolithencave consist of dolomite. Due to more dissolved ions





in the drip water and as well still air filled cavities, more PCP takes place. This results in an
increase of both the total amount dissolved $Ca^{2+}$ ions and the Mg/Ca ratio of the drip water.
Consequently, during years of higher rainfall, both growth rate and the annual Mg content of
the speleothem should increase (Wassenburg et al., 2012). Thus, both growth rate and mean
annual Mg concentration should be proxies for past precipitation. The opposite interpretation
of the Mg concentration on the seasonal and annual time-scale highlights the complexity of
trace element signals in speleothems. A similar phenomenon was also detected by Treble et
al. (2003) for Sr.

*5.3.2 UV-luminescence*
Luminescence in speleothems induced by UV-light has been associated due to humic and
fulvic acids, which are transported from the soil zone into the cave via the drip water
(McGarry and Baker, 2000; van Beynen et al., 2001; Shopov, 2003). In the UV-microscopy
picture, it is obvious that the brownish layers are luminescent, which is due to their higher
content of humic and fulvic acids originating from the soil. These layers are formed during
autumn and winter, when the organic material produced during the vegetation period is
flushed into the cave (Sundqvist et al., 2005; Orland et al., 2012).
The G/B ratios taken with the *Aavatech* core scanner have been interpreted as reflecting the
amount of humic acids in corals (Grove et al., 2010). This relationship should generally be
also valid for speleothems. Since, Y and P are elevated in the brownish layers (Borsato et al.,
2007), a positive correlation between the G/B ratio and these elements should occur. Indeed,
Y has been associated with fluorescent laminae in speleothems (Fairchild et al., 2010). This is
not the case for the three Zoo-rez stalagmites. The reason for this observation is the inclusion
of detrital material in all stalagmites, which shows no or only very low UV-luminescence.
This is obvious in the comparison of the G/B ratio with the content of Mn and Y (Fig. 13).
The G/B ratio is low when Mn and, therefore, the content of detrital material is high. In some
sections Y is positively correlated with the G/B ratio. However, in other sections, containing
more detrital material (which may also contain Y), a negative correlation is observed to the
G/B ratio. In this case, the humic acid signal in the G/B ratio is overprinted by detrital
material. Similarly, the long-term decreasing trend in the G/B ratios of the three Zoo-rez
stalagmites results from the higher amount of detrital material in the top sections of the
stalagmites and cannot be used as a proxy for past precipitation. In summary, the G/B ratio





472   appears to be not appropriate to detect changes in humic acid content of these speleothems,

473   which contain – at least in some parts – relatively high amounts of detrital material.

474   Furthermore, as discussed previously, the resolution of 70 µm/pixel makes it impossible to

475   detect thinner (i.e., with a thickness < 140 µm) annual laminae.

476

477   *5.3.3 Lamina thickness*

478   We interpret annual lamina thickness as a proxy for past precipitation. Thus, we correlated the

479   lamina thickness series to instrumental data from the meteorological station Bamberg

480   (www.dwd.de), which provides data from 1949 AD to present. Since stalagmite Zoo-rez

481   stopped growing in 1970 AD, only 22 years of overlapping proxy and instrumental data are

482   available. We found no significant correlation between lamina thickness and surface

483   temperature, neither for the annual mean nor for individual months. In contrast, a positive, but

484   insignificant correlation ($r = 0.33$; $p > 0.05$; $N = 22$) between lamina thickness and annual

485   precipitation was found, as has been reported in other studies (Genty and Quinif, 1996;

486   Proctor et al., 2000). In order to test for a better correlation, the lamina thickness chronology

487   was shifted along the precipitation time series (both back in time and up to 1999 AD, the year

488   when the stalagmites were sampled). The maximum correlation was found, assuming a

489   cessation of stalagmite growth in 1970 AD. This is in good agreement with the dating results

490   (see section 5.1). A probable reason for the three stalagmites to stop growing in 1970 could be

491   that further exploration of the deeper parts of the cave started in 1971. Furthermore, the

492   correlation between precipitation of all individual months of the previous, the current, and the

493   next year and lamina thickness was calculated (cf., Tan et al., 2006). In addition, different

494   seasons were compiled to check whether the correlation increases. This is a standard approach

495   in tree-ring research (e.g., Treydte et al., 2001; Buentgen et al., 2005; Wilson et al., 2005;

496   Konter et al., 2014). The highest correlation for an individual month ($r = 0.64$; $p < 0.001$) is

497   observed for December of the current year and $r = 0.57$ ($p < 0.01$) is observed for the season

498   of previous December and current January, February, March, April, May, and December

499   (DJFMAMD). Most probably the clear layer is formed during January to May and the

500   brownish layer during December. This would also explain the contribution of the previous

501   December, because the boundary from the brownish to the clear layer is not sharp (Fig. 4a)

502   and could be not exactly at the end of one year. This shows that stalagmite growth is

503   dominated by the winter season as expected from the higher amount of recharge during winter



(Wackerbarth et al., 2010; Mischel et al., 2015). Furthermore, this proves that the upper limit of the brownish, pigmented layers corresponds to the end of the year.

These results provide the background in order to reconstruct past precipitation further back in time using lamina thickness in speleothems from Zoolithencave. A comparison of the sum of precipitation during DJFMAMD and the lamina thickness series with a precipitation reconstruction based on tree-ring width in the Bavarian forest (Wilson et al., 2005) shows a similar pattern (Fig. 15) further supporting that lamina thickness reflects past precipitation variability. Note that the tree-ring width reconstruction is for spring and summer months (March, April, May, June, July, and August; MAMJJA) and, thus, a different season than our record.

## 6   Conclusions

1. The arrangement of the three Zoo-rez stalagmites grew from 1800 to 1970 AD, which is supported by the detection of the $^{14}$C bomb peak, $^{14}$C-dating of a charcoal piece below the stalagmite, and lamina counting.

2. The three stalagmites show three types of annual laminae: visible, UV-luminescent, and elemental laminae.

3. Visible laminae consist of a clear and a brownish pigmented layer pair. Measurements of lamina thickness along five tracks on the three stalagmites results in a cross-dated lamina thickness chronology, which is a proxy for winter and spring (DJFMAMD) precipitation.

4. UV-luminescent laminae correspond to the brownish pigmented layers. Using UV-luminescence scanning, the annual laminae could not be detected due to the minimum resolution of 70 µm/pixel.

5. Elemental laminae are clearly visible in Mg, Ba, and Sr, and are strongest for Mg. All three elements are influenced by PCP, which is higher during spring and summer and lower during autumn and winter.

6. Yttrium and P content are higher in the brownish pigmented layers and induced by an annual flush of humic and fulvic acids, when infiltration increases. Both elements are also incorporated in association with detrital material. Thus, Y and P are no clear precipitation proxies. Manganese and Al are associated with detrital material.



7. A mean curve of the Mg content of all tracks was wiggle matched to the lamina thickness chronology resulting in an annual Mg time series. This correlates positively with the lamina thickness chronology and is also influenced by PCP, which is higher in years with more precipitation due to more active vegetation and therefore, more hostrock dissolution.

8. These results highlights the potential of annually laminated speleothems from Zoolithencave for reconstruction of past precipitation variability.

## Acknowledgments

Dana Riechelmann is grateful to the DFG for funding (RI 2136/2-1). We thank the "Forschungsgruppe Höhlen und Karst Franken e.V." for the permission to sample stalagmites from Zoolithencave and support during the cave trip in 1999. For assistance during LA-ICPMS, we thank B. Stoll and U. Weis from MPI for Chemistry, Mainz (MPIC) as well as J. Faust. Furthermore, we thank B. Schwager (MPIC) for support during sample preparation for [230]Th/U-dating. UV-Luminescence measurements were supported by B. Koster at the NIOZ. We also thank K. Seelos for scanning the thin sections and M. Veicht for some graphic editing. B. Kromer from the Curt-Engelhorn-Zentrum for Archeometry gGmbH (Mannheim) is thanked for [14]C-analyses. Finally, we thanks A. Immenhauser for the possibility to use the UV-fluorescence microscopy facility at the Ruhr-University Bochum.

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



Table 1. Results of $^{230}$Th/U-dating. All errors are given at the 2σ-level.

| Sample | dft | $^{238}$U [µg/g] | $^{234/238}$U | $^{230}$Th/$^{238}$U | $^{230/232}$Th | Age uncorr. [ka] | Age corr. [ka] |
|---|---|---|---|---|---|---|---|
| Zoo-rez-1-o | 0.1-0.6 cm | 0.0351 ±0.0002 | 1.1686 ±0.0046 | 0.0695 ±0.0016 | 2.6 ±0.1 | 6.6822 ±0.1629 | 4.6699 ±0.9998 |
| Zoo-rez-1-u | 1.2-1.7 cm | 0.0233 ±0.0001 | 1.1881 ±0.0052 | 0.0639 ±0.0031 | 0.9 ±0.04 | 6.0255 ±0.2985 | 0.3402 +3.3143 -0.2947 |




















Table 2. Results of [14]C-dating of charcoal and carbonate.

| Sample | Age [a] refer to 1950 AD | $\delta^{13}$C [‰] | [14]C-activity [pmC] | Cal 1$\sigma$ |
|---|---|---|---|---|
| Zoo-rez-1, 0.8 mm dft | -1,440 ± 22 | -10.5 ± 0.3 | 119.6287 ± 0.32494 | |
| Zoo-rez-1, 2.2 mm dft | -527 ± 22 | -8.6 ± 0.3 | 106.7865 ± 0.297579 | |
| Zoo-rez-1, 6.9 mm dft | 740 ± 24 | -7,4 ± 0.3 | 91.19406 ± 0.275865 | |
| Zoo-rez-2, charcoal | 165 ± 21 | -23.0 ± 2 | | AD 1671-1951 |




















Table 3. Correlation coefficients calculated between the different elemental concentrations of
the individual tracks. a) Zoo-rez-1.1, b) Zoo-rez-1.2, c) Zoo-rez-1.3, d) Zoo-rez-2, e) Zoo-rez-
3. Correlation coefficients, r > 0.25 are marked in green, r > 0.5 in orange, and r > 0.7 in red.
Negative correlation coefficients, r < -0.3 are marked in blue. All coloured correlations have p
values < 0.001.

| a) | Mg | Al | P | Mn | Sr | Y |
|---|---|---|---|---|---|---|
| Mg | | | | | | |
| Al | 0.10 | | | | | |
| P | 0.07 | 0.25 | | | | |
| Mn | 0.22 | 0.45 | 0.47 | | | |
| Sr | 0.19 | 0.03 | -0.01 | 0.14 | | |
| Y | -0.13 | 0.24 | 0.32 | 0.44 | 0.01 | |
| Ba | 0.32 | 0.05 | 0.04 | 0.14 | 0.30 | 0.06 |


| b) | Mg | Al | P | Mn | Sr | Y |
|---|---|---|---|---|---|---|
| Mg | | | | | | |
| Al | 0.12 | | | | | |
| P | 0.14 | 0.21 | | | | |
| Mn | 0.18 | 0.31 | 0.65 | | | |
| Sr | 0.16 | -0.01 | -0.03 | 0.02 | | |
| Y | 0.07 | 0.07 | 0.52 | 0.51 | 0.07 | |
| Ba | 0.19 | 0.02 | 0.04 | 0.06 | 0.18 | 0.10 |


| c) | Mg | Al | P | Mn | Sr | Y |
|---|---|---|---|---|---|---|
| Mg | | | | | | |
| Al | 0.22 | | | | | |
| P | 0.12 | 0.26 | | | | |
| Mn | 0.23 | 0.45 | 0.78 | | | |
| Sr | 0.19 | 0.02 | -0.01 | 0.02 | | |
| Y | 0.02 | 0.26 | 0.68 | 0.72 | 0.04 | |
| Ba | 0.35 | 0.07 | 0.12 | 0.17 | 0.27 | 0.16 |


| d) | Mg | Al | P | Mn | Sr | Y |
|---|---|---|---|---|---|---|
| Mg | | | | | | |
| Al | 0.09 | | | | | |
| P | -0.09 | 0.23 | | | | |
| Mn | 0.18 | 0.26 | 0.28 | | | |
| Sr | 0.33 | 0.01 | -0.07 | 0.03 | | |
| Y | -0.46 | 0.06 | 0.47 | -0.02 | -0.17 | |
| Ba | 0.63 | 0.17 | 0.09 | 0.13 | 0.40 | -0.10 |


| e) | Mg | Al | P | Mn | Sr | Y |
|---|---|---|---|---|---|---|
| Mg | | | | | | |
| Al | 0.10 | | | | | |
| P | 0.09 | 0.13 | | | | |
| Mn | 0.14 | 0.24 | 0.34 | | | |
| Sr | 0.17 | -0.03 | -0.03 | 0.05 | | |
| Y | -0.12 | 0.09 | 0.45 | 0.22 | -0.06 | |
| Ba | 0.51 | 0.08 | 0.21 | 0.24 | 0.18 | 0.11 |





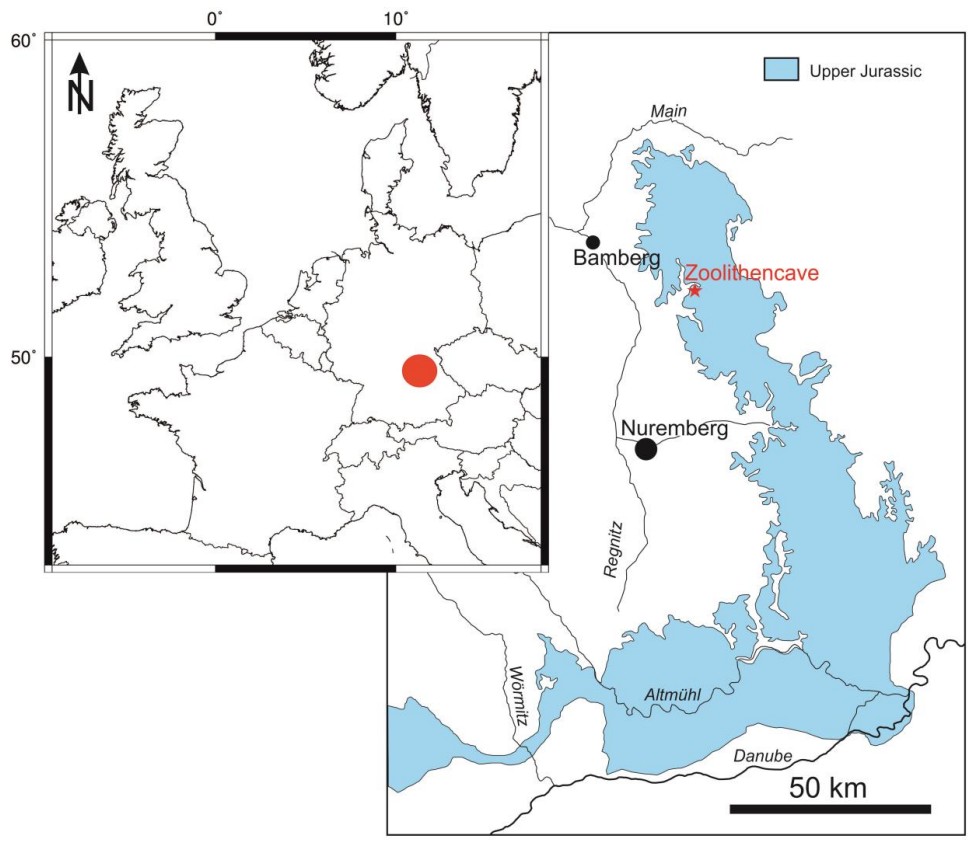


Figure 1. Map of the Upper Jurassic containing marl, limestone, and dolomite (modified after
Groiß, 1988). The location of Zoolithencave is indicated by the red star. Location of the
region is marked in the map of Central Europe in the upper left.












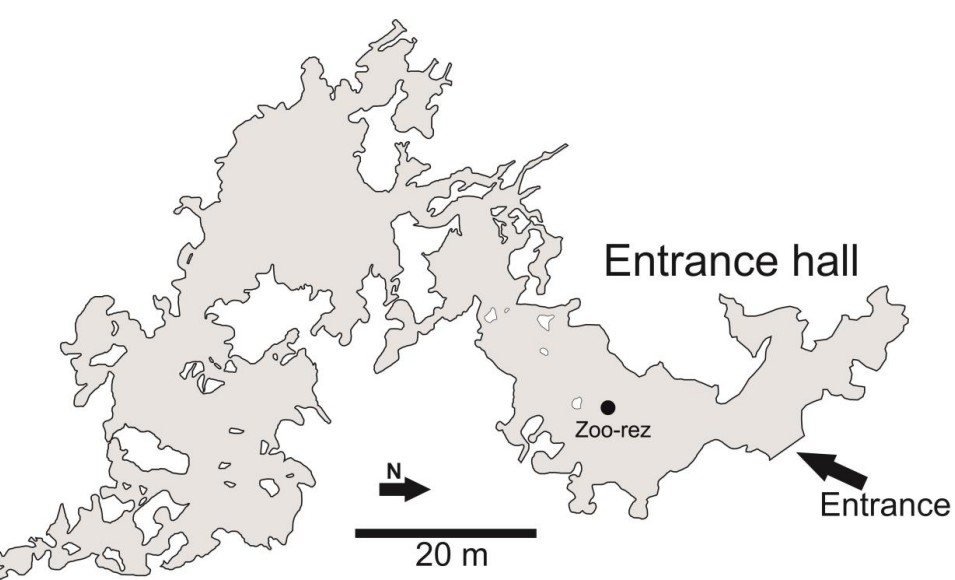


Figure 2. Map of Zoolithencave with the sampling site of Zoo-rez indicated (modified after
Dreyer, 2000).







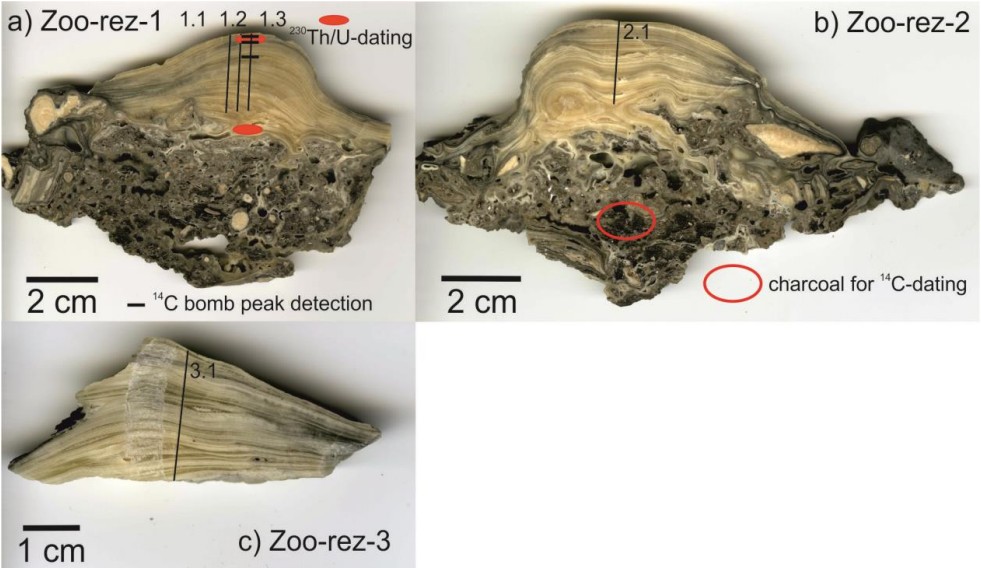

Figure 3. Pictures of the sampling slices of Zoo-rez-1 (a), Zoo-rez-2 (b), and Zoo-rez-3 (c) subsequent to cutting. The laser ablation tracks (labelled 1.1, 1.2, 1.3, 2.1, and 3.1, respectively), the sampling positions for $^{14}$C bomb peak detection as well as $^{230}$Th/U-dating, and the charcoal used for $^{14}$C-dating are indicated.



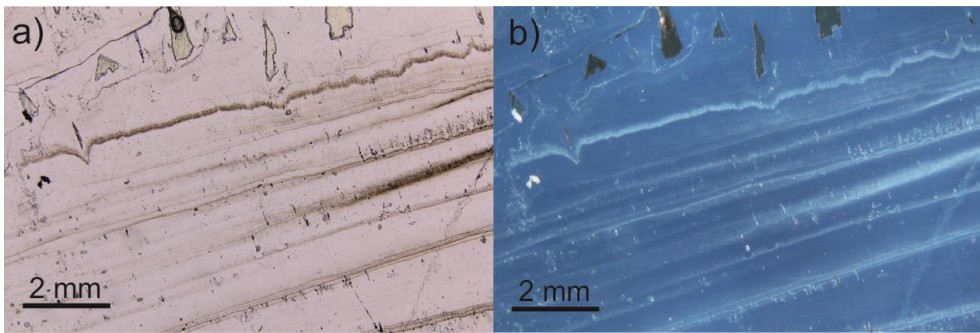

Figure 4. a) Visible laminae in Zoo-rez. Which is present as layer pairs consisting of a clear and a brownish pigmented layer. b) Fluorescence is stronger for the brownish pigmented layers than for the clear layers.




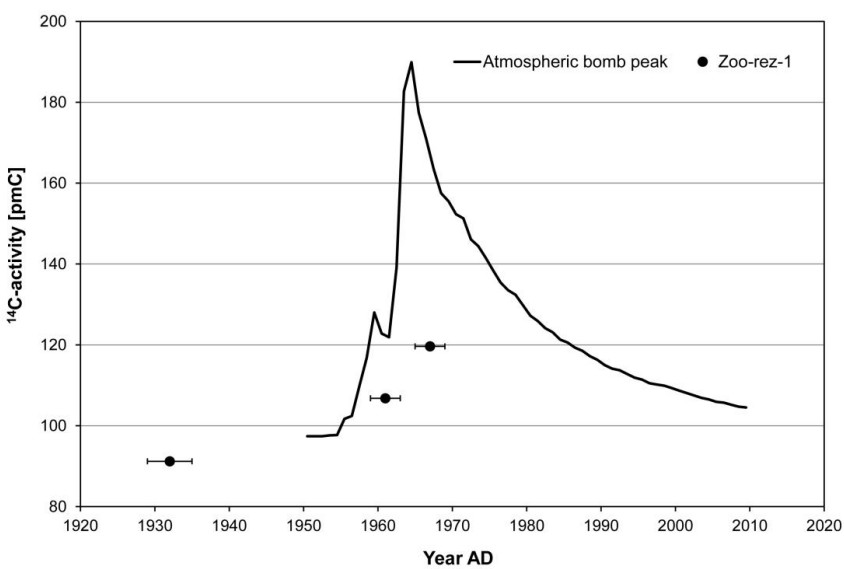


Figure 5. $^{14}$C-activity determined for three samples from the top section of Zoo-rez-1 (at 0.8,
2.2, and 6.9 mm dft, respectively) compared with the atmospheric bomb peak (Hua et al.,

961  2013).

















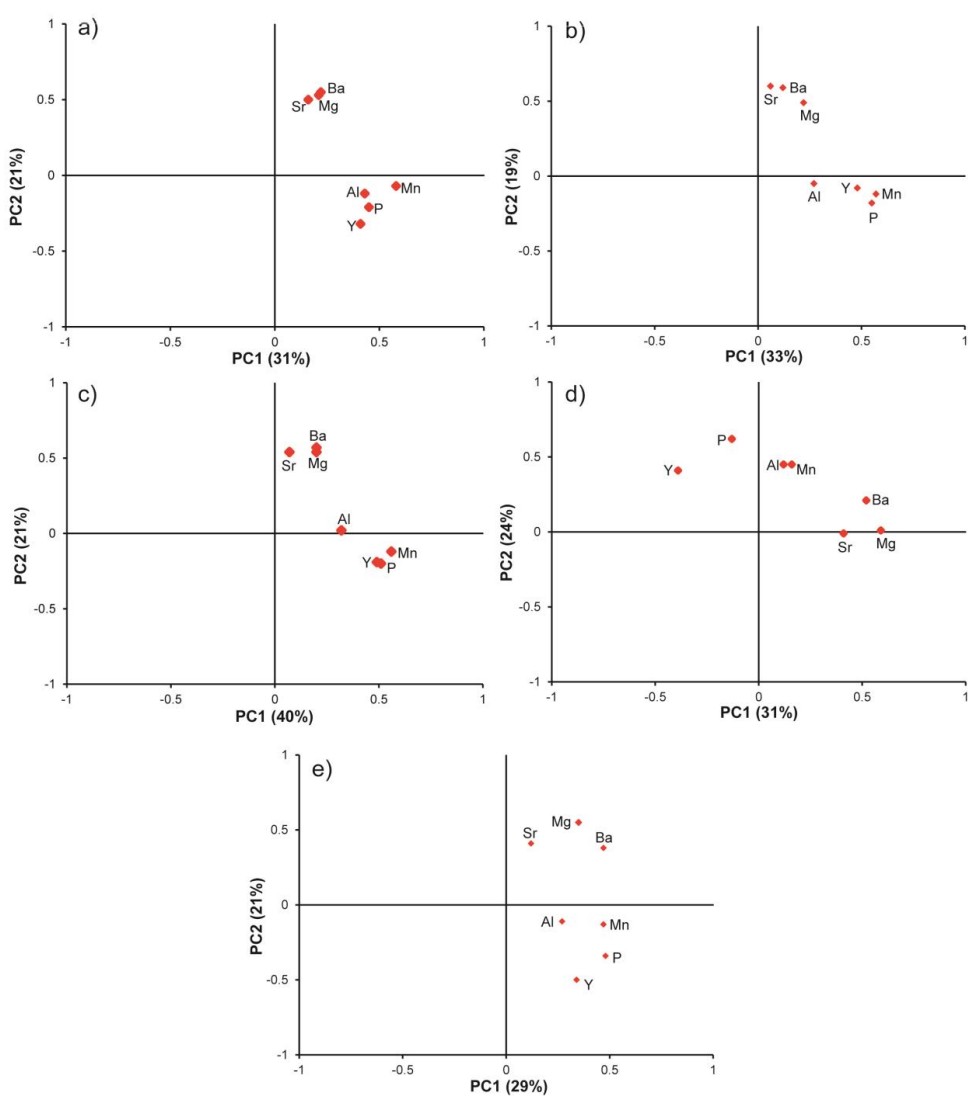


Figure 6. Results of PCA of the element data for the five different tracks. Zoo-rez-1.1 (a),
Zoo-rez-1.2 (b), Zoo-rez-1.3 (c), Zoo-rez-2 (d), and Zoo-rez-3 (e).





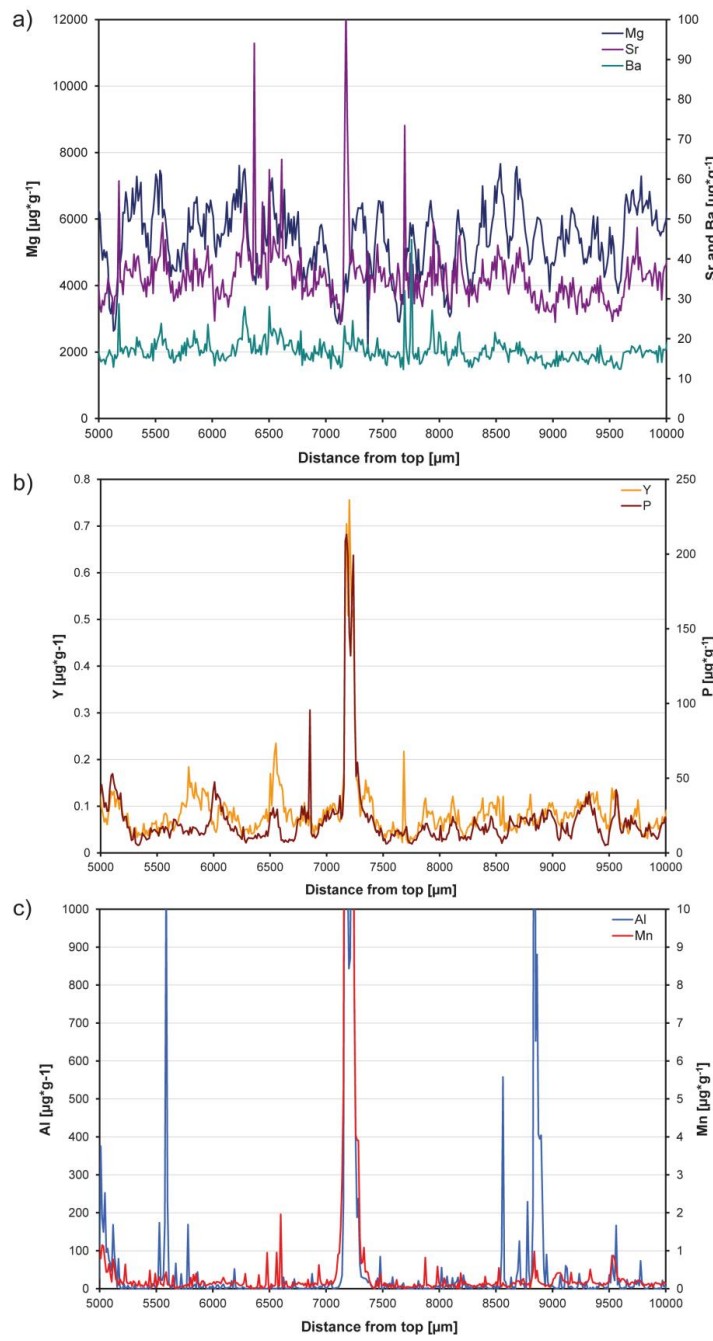


Figure 7. Compilation of Mg, Ba, and Sr (a), Y and P (b), and Al and Mn (c) concentrations in
Zoo-rez-1.1 in section 5000 to 10,000 μm dft. The same patterns are observed for Zoo-rez-
1.2, 1.3, 2, and 3.





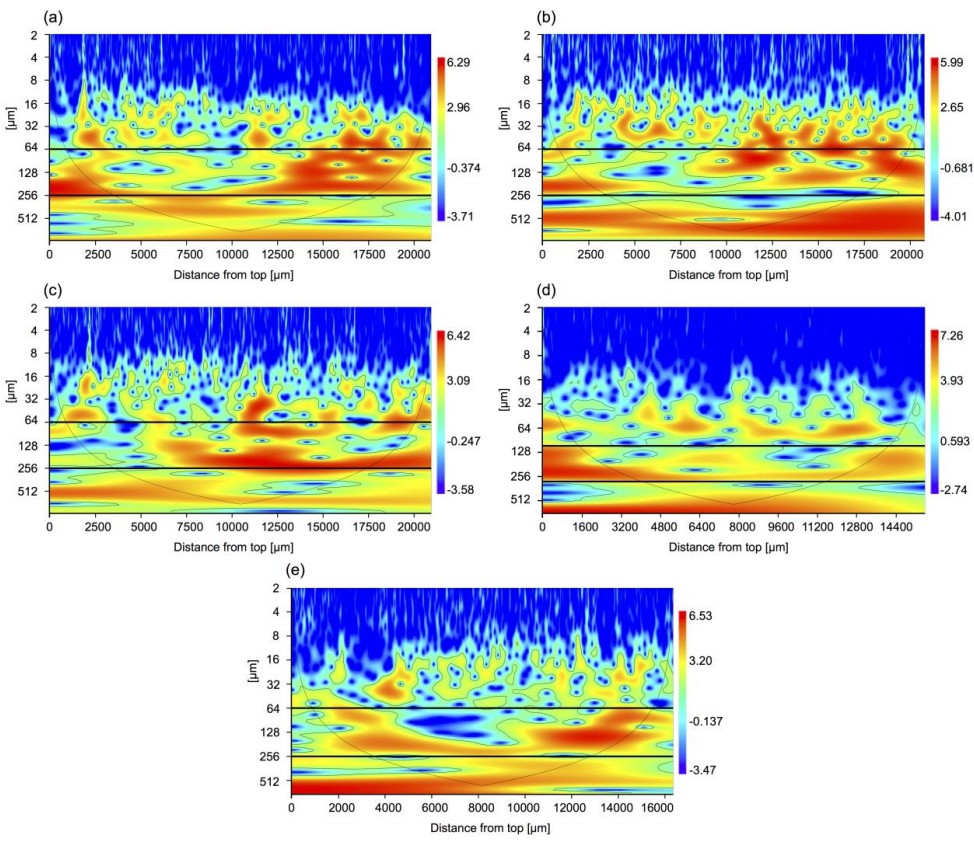


Figure 8. Wavelet analysis of the Mg concentration of the five tracks: Zoo-rez-1.1 (a), Zoo-
rez-1.2 (b), Zoo-rez-1.3 (c), Zoo-rez-2 (d), and Zoo-rez-3 (e).











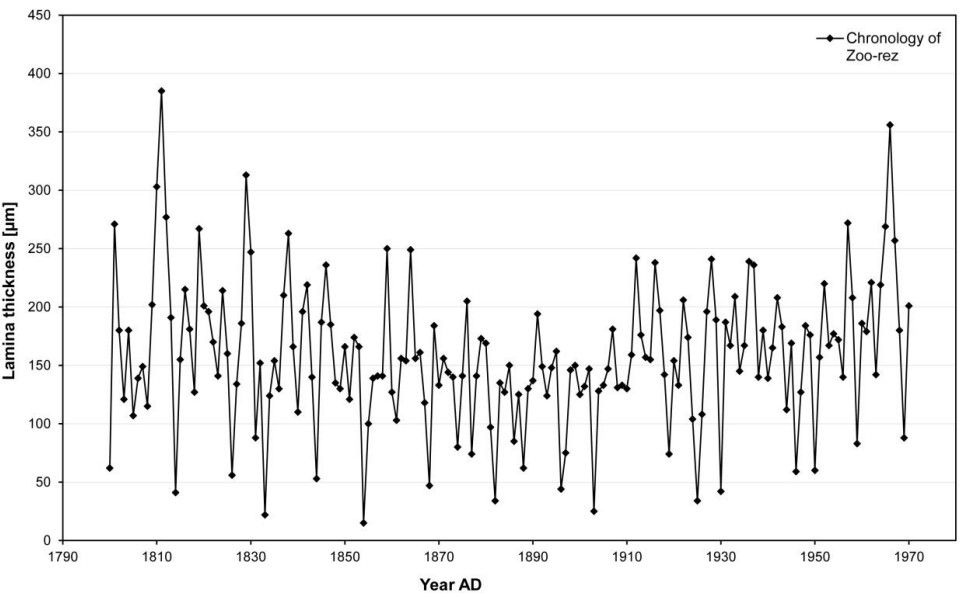


Figure 9. Mean lamina thickness chronology of all five tracks measured on stalagmite Zoo-

rez.

















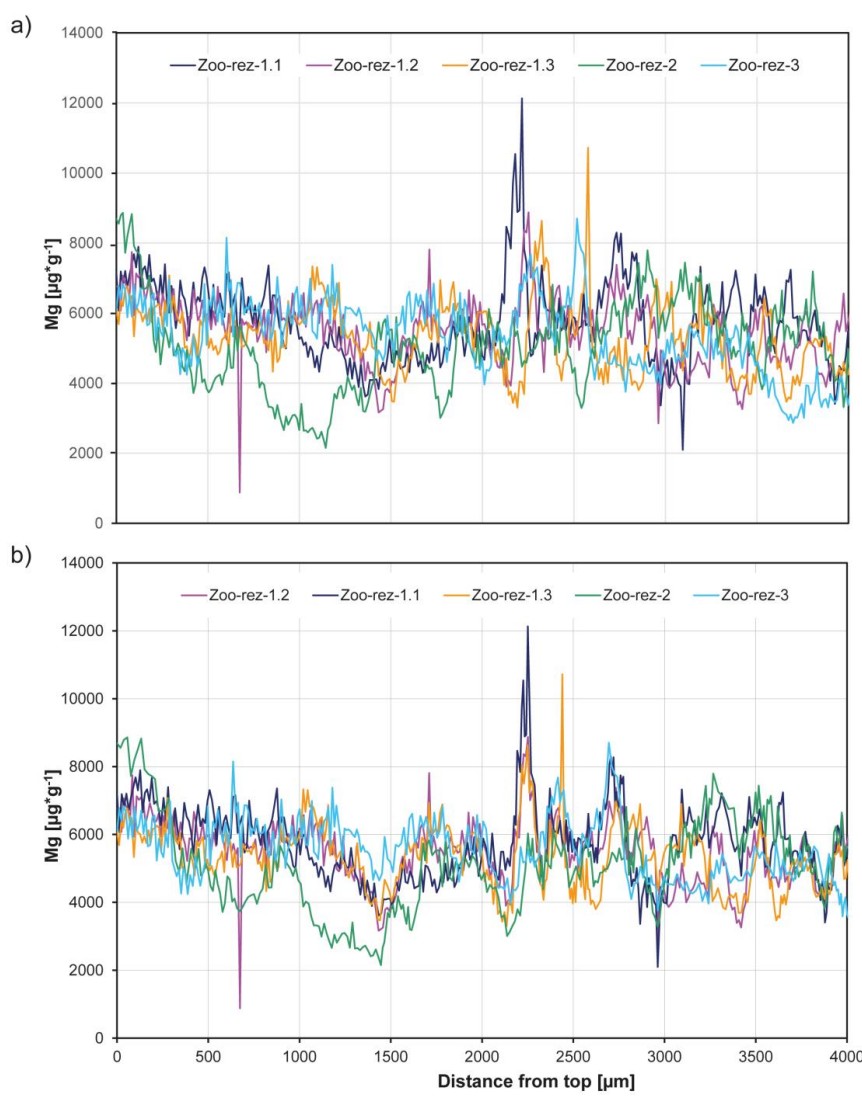


Figure 10. Mg concentration along the five individual tracks on stalagmite Zoo-rez in the
section 0 to 4000 μm dft before (a) and after (b) wiggle matching.







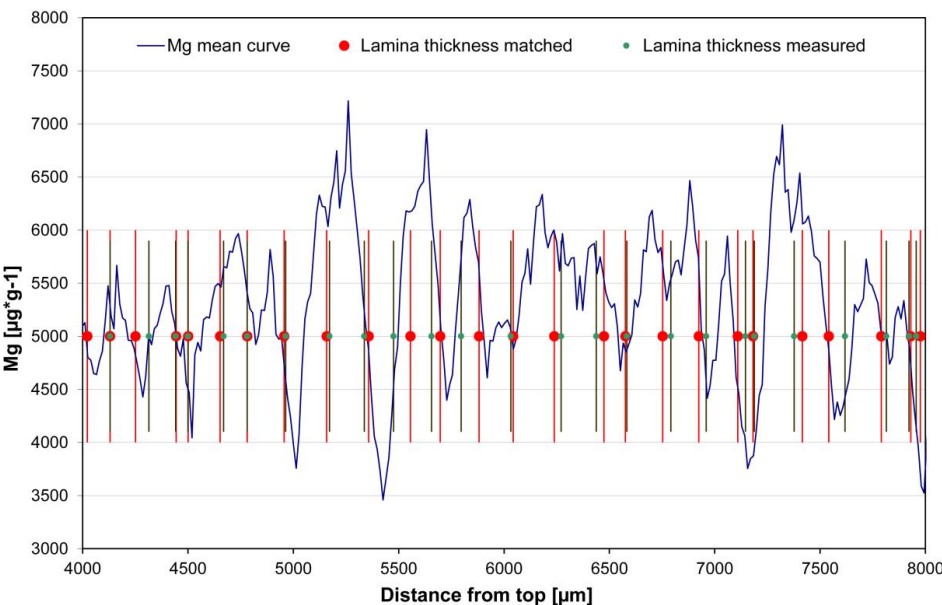


Figure 11. Mean curve of Mg concentration compared with both the matched (red lines) and
the measured (green lines) lamina thickness series for the section between 4000 and 8000 μm
dft. The boundaries of the lamina were matched to the increase (from bottom to top of the
stalagmite) of the Mg concentration.










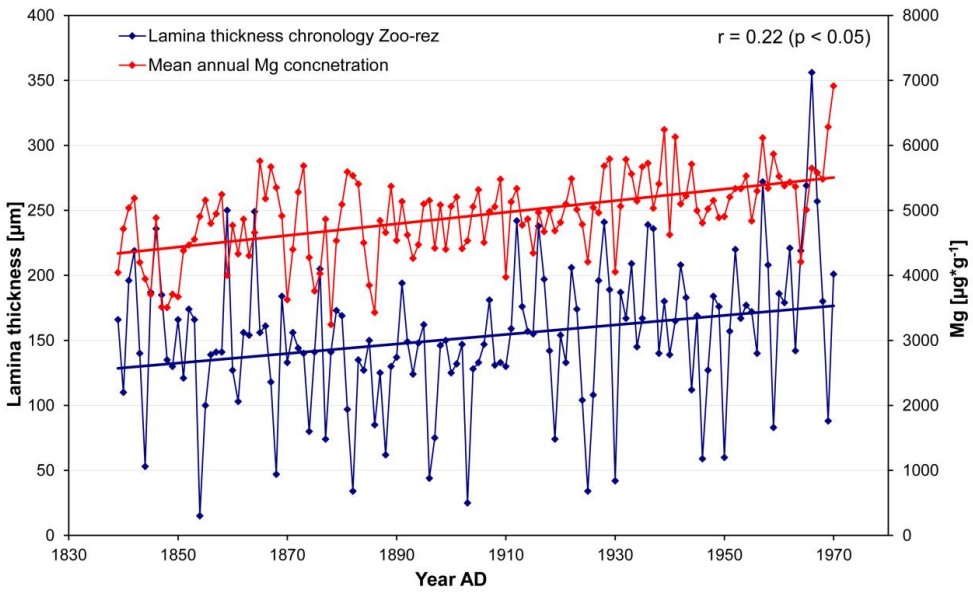


Figure 12. Comparison of lamina thickness and mean annual Mg concentration. The
correlation coefficient is r = 0.22. The straight lines represent linear fits of the time series.















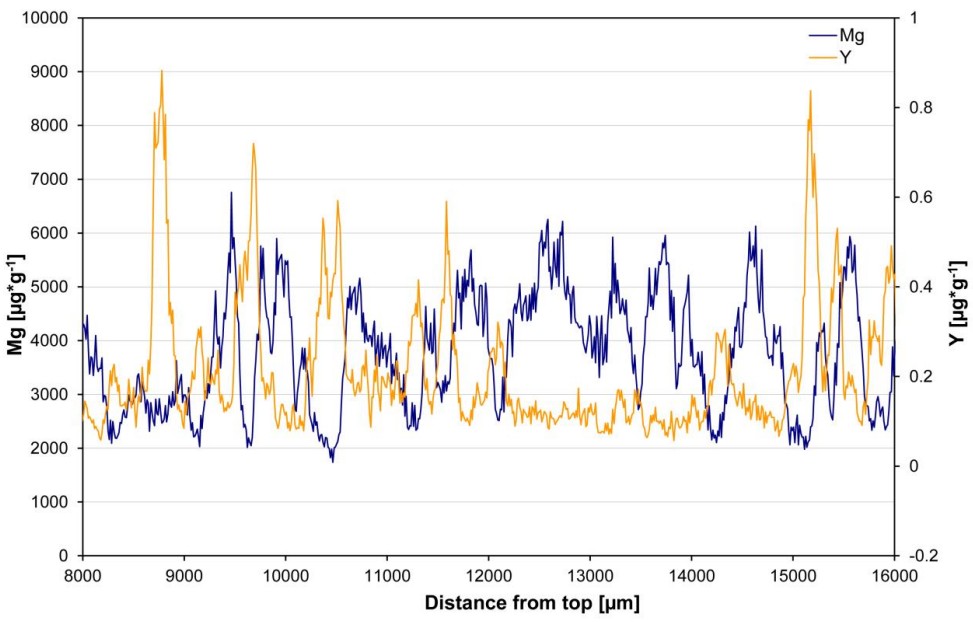


Figure 13. Evolution of Mg and Y on track Zoo-rez-2 between 8000 and16,000 μm dft.


















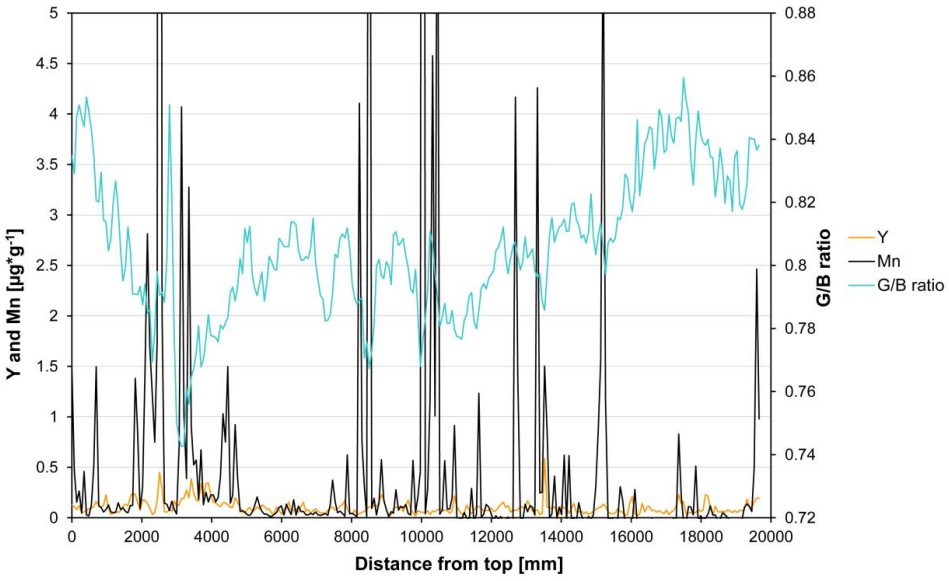


Figure 14. Comparison of the G/B ratio with the content of Y and Mn for track Zoo-rez-3.

















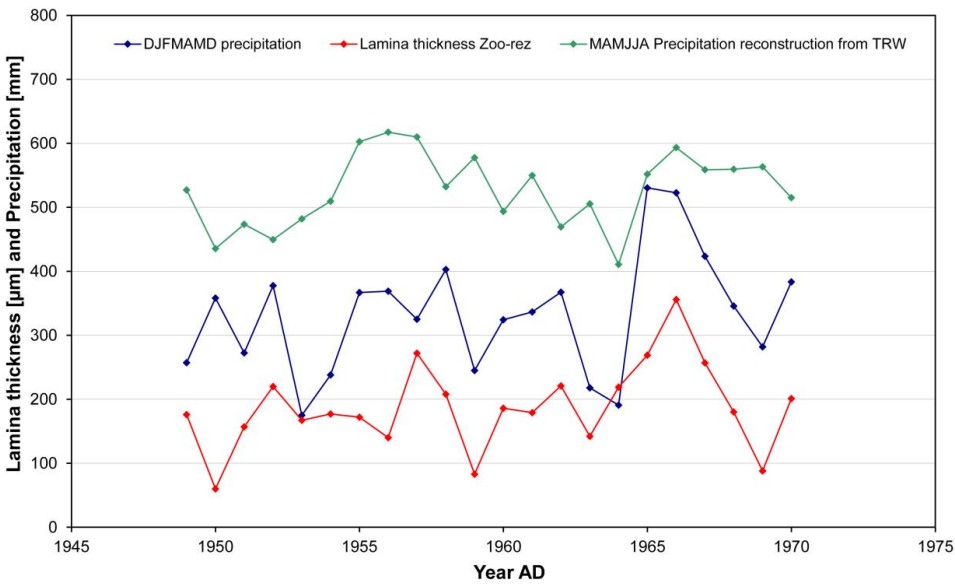


Figure 15. Comparison of the sum of the amount of precipitation during previous December,
January, February, March, April, May and December (DJFMAMD) at the meteorological
station Bamberg (DWD), a precipitation reconstruction for March, April, Mai, June, July, and
August (MAMJJA) based on tree-ring width (Wilson et al., 2005) and the lamina thickness
chronology of Zoo-rez.









