# Peer review of "Detection and origin of different types of annual laminae in"

_Climate of the Past, 2016_

## Short Comment (SC1) · 16 Feb 2016

While the annual lamina classification is very interesting and, if I'm accurate, first of its kind, Class V is designated as 'mineralogical' while I would expend it to petrographic or include classification VI, follow Frisia et al., 2000 and onwards.

Also, a more updated reference for fluorescence seasonal lamination could be Orland et al., 201(Chem. Geo.)
[Figure]

Very nice paper,

Good luck.

---

## Short Comment (SC2) · 16 Feb 2016

Sorry, forgot to add line # - lines 69-70
* * *

---

## Referee Comment (RC1) · Anonymous Referee #1 · 12 Apr 2016

Some general comments:

The authors present a study of modern stalagmites from an intersting cave site in Germany also known for its archeological and paleontological remains. Studying various types of lamination patterns in speleothems and other paleoclimate archives is undoubtedly important in order to establish robust high-resolution connections to climate relevant variables. In this regard the manuscript is relevant to various climate studies published in journals as "Climate of the Past".

[Figure]

In the manuscript a comparison of petrographic and elemental evidence to meteorological data is provided. In its current form, however, the results and discussion are not really bridging the gap from proxy data to instrumental meteorological and numeric climate data. Envisaging a "quantitative reconstruction" based on proxy evidence is always challenging and with regard to speleothems might only be reliable in connection with a deeper understanding of the actual cave/speleothem vs. climate relationships, i.e. also based on results from high-resolution cave monitoring. Such data and relationships are not presented in the manuscript and consequently some of the explanations given are still of some qualitative character.

The stalagmite samples presented are difficult with regard to establishing a U-Th based chronology (due to detrital contamination). An approach at its best is provided via an indirect age constraint based on radiocarbon dating of charcoal and partially detecting the evolution of 14C bomb peaks. Altogether, however, the chronology (being a critical issue regarding the intention of this study) is not robust in a stricter sense making the annual nature of the lamination and some of the relations discussed a subject to debate.

Of interest and high relevance in this study are the various statistical methods of variable complexity being applied to the datasets, e.g. methods typically applied in tree-ring research. Although all of these methods are well-known and might be (widely) accepted, the critical reader encounters some flavor of significant and sometimes subjective (e.g. inserting specific numbers of laminae, timing of 14C bomb peak evolution) tuning being applied. Some scientists will consider this problematic regarding the reliability of the actual proxy-climate relationships and there are controversial and ongoing discussions on that topic. In some speleothem records the connections proposed are clearly more obvious, i.e. without sophisticated statistical treatment.

Considering the origin and rhythm of the lamination observed and some of the parameter relationships the discussion should me more critical and diverse and also open to possible alternative explanations. Some of the conclusions drawn seem too simple

based on the chronology and data presented.

Particular chapters/paragraphs/lines:

Introduction: Informative and comprehensive, e.g. regarding the broad literature cited.

Line 60ff: Regarding the various lamina types the frequent occurrence of "event lamination" could also be mentioned, e.g. lamination resulting from flooding of cave chambers (clay layers) being a potential proxy for high rainfall and floods in continental areas or lamination in some coastal caves representing sea-level fluctuations (e.g. Antonioli et al., 2004; Dorale et al., 2010).

Dating methods: Lines 131f: Why do the authors not use the updated half-lives reported in Cheng et al. (2013, EPSL)?

No U-Th data are available from stalagmite Zoo-rez-2. Maybe this stalagmite is significantly cleaner (less detrital contamination) and/or the U contents are higher. The petrography looks somewhat lighter in color (Fig. 3), so the stalagmite might be cleaner eventually. Might be worth trying.

Visible laminae: This is a critical issue. Based on the microscopic pictures shown (Fig. 4; 2 mm scale bar!) some of the clear (annual) layers look much thicker (up to mm-scale) than supported by the average lamina thickness calculations, i.e. the visual inspection is not consistent with the calculated thicknesses (few hundreds of microns). Sometimes brownish layers are very close to each other and even look like very thin doubled brownish layers (cf. Baker et al., 2002) followed by thick clear layers. Based on visual inspection the lamina development looks quite unregular in particular with regard to lamina thickness and it could even be some kind of event lamination.

Considering the absolute lamina counts from the parallel tracks of Zoo-rez-1 (124 vs. 161 vs. 135; lines 201ff) these (small) numbers are relatively far from each other despite their proximity within the same stalagmite. This should be addressed more critically.

Suggesting visible annual laminae the authors should provide some more direct (microscopic) comparison of, e.g. petrography vs. selected trace elements within a typical stalagmite section. This is frequently provided in speleothem lamination studies and gives more confidence regarding high-resolution correlations.

Chronology of stalagmite Zoo-rez: The 14C-dating of a basal charcoal piece might be considered a minimum age constraint but does not necessarily constrain the absolute age of initial stalagmite formation – in particular when reviewing the position of the charcoal piece, i.e. significantly below initial stalagmite growth (cf. Fig. 3). A significant amount of sedimentation time might have passed between 14C age and start of lamination development and therefore it is not a direct age constraint compared to U-Th data.

A calibrated 1-sigma dating range of 1671-1951 AD is reported for the single radiocarbon age (line 230). This is a relatively large range and therefore only moderate constraint of the initial growth. It should be addressed more critically.

Regarding the detection of the 14C bomb peak(s): Information on radiocarbon in speleothems is given well. In order to capture the timing of decrease in atmospheric radiocarbon activity, however, some additional 14C data would be needed to (better) constrain the timing of growth cessation. "Since the increase in 14C in Zoo-rez is large (compare e.g., Noronha et al., 2015), the peak is near the maximum of the atmospheric 14C values." (lines 243f). Based on the data points presented in Fig. 5 this seems rather speculative and in fact not a strong anchor for the claimed (narrow) timing of the stalagmite top (1970 AD +- 5 years; cf. lines 245ff). The upper limit of 14C activity in drip water from this cave site might be significanlty higher before decreasing to lower values.

"Zoo-rez did not grow until 1999 AD" (year of sampling; cf. lines 239f). Instead, growth cessation in 1970 AD is suggested. This is strange considering that the stalagmite was fed by an active drip suggesting recent growth (lines 120f). Why is the drip water not

supersaturated? Why is the system potentially disturbed?

Elemental laminae A comparison of the modern stalagmite composition with modern hydrochemical data would be of high value in order to establish robust relationships, e.g. annual variability of elemental supply.

Procedure and interpretation of wavelet analysis shown in Fig. 8: Provide more explanation!

lines 270ff: The correspondance of Mg concentration and lamina thickness does not prove the annual nature of the lamination pattern, only some possible interrelation between the two parameters. However, this interrelation should also be supported by some more direct comparison.

Al and Mn concentrations are considered proxies for detrital content in the stalagmites. What about silicon? This element is most typical for siliciclastic input, e.g. clay minerals, feldspars.

Interpretation and Discussion

Chronology lines 300ff: A corresponding number of missing laminae was inserted into the chronology. 124 vs. 161 counted laminae constitute a significant mismatch. Moreover, lamina thickness was assumed for insertion. So, nine laminae were inserted into the "master track". The reader gets an impression of playing around, i.e. a semi-quantitative and somewhat arbitrary approach. Provide more information and argumentation with regard to this procedure!

lines 315ff: Please explain meaning and significance of the correlation procedure and values!

lines 327ff: Regarding the lamina adaptions and radiocarbon age constraints: See comments above. Altogether it seems a somewhat floating chronology.

lines 332: A critical issue. Cave ventilation is excluded by the authors regarding lamina development and measured (monitored) cave pCO2 values. However, an annual amplitude of around 1000 ppmv in pCO2 is relatively large, i.e. more than twice the absolute atmospheric concentration. Moreover, the obviously higher values are "removed" seasonally approaching atmospheric compositions during wintertime. Also, the stalagmites were collected from the entrance section being more prone to influences of cave air exchange. Consequently, there are several major arguments for cave ventilation playing a potential role. Excluding such effects in your interpretation is too simple in this particular case. The authors further state: "...stalagmite growth is dominated by the winter season" (lines 502f). This is frequently the case in seasonally ventilated cave systems of the mid latitudes, i.e. another argument for cave ventilation most likely playing a significant role for speleothem deposition.

lines 337f: Annual lamination in stalagmite Zoo-rez is related to annual changes in drip water composition. Give more information and explanation! What kind of changes? Are monitoring data available/published?

Fig. 9 shows a strong variation in annual lamina thickness. This looks interesting. Is there some multi-annual cyclicity represented?

Wiggle matching and data interpolation lines 357f: The shifts performed during the wiggle matching procedure are up to several typical lamina-thicknesses. So, particular numbers of laminae are inserted and subsequently significantly shifted. This is critical regarding a robust chronology and data interpretation.

lines 366f: "The clear layer corresponds to higher Mg concentration and the brownish pigmented to lower Mg concentration." This seems mainly based on wiggle matching. It should be shown easily by a direct comparison of Mg concentration vs. stalagmite petrography on a microscopic scale (visually), i.e. the resolution is high enough in both proxies.

Fig. 12: After sophisticated tuning treatments a direct comparison from year to year is still difficult. There is an increasing trend in lamina thickness and Mg – this is interesting

and might be worth of further discussion.

Interpretation of proxy signals in terms of past climate variability lines 385ff: As already mentioned, the seasonal relations proposed would be more robust in combination with some hydrochemical data.

Fig. 13: Some anticorrelation of Mg and Y. This deserves more discussion/explanation.

Does some positive correlation of increased Sr, Ba with thicker laminae exist? Some studies relate the Sr, Ba incorporation to carbonate precipitation rates (e.g. Lorens, 1981; Tesoriero and Pankow, 1996; Huang and Fairchild, 2001).

UV-luminescence The brownish layers probably depend on humic- and fulvic acids from the soil (constraint from UV luminescence). However, these components could also be introduced during snowmelt flushing events. As mentioned, in the microscopic image some of the brownish layers look like a double-layer potentially respresenting two events close to each other, e.g. snowmelts.

line 464: Fig. 14 (not 13).

lines 465: The authors talk about "detrital" material/contamination. What kind of detritus? Being siliciclastic material Si and Al should be typically high. If Mn is high then I would also expect Fe being relatively high. The form and origin of detrital contamination is not constrained well and therefore somewhat speculative.

Lamina thickness Fig. 15: Some correspondance but also some mismatch (earlier part).

References: Informative and extensive. A broad range of the literature available on that topic is represented.

Finally: Considering the interesting and paleoclimatically relevant topic, as well as broad approach involving various analytical and statistical techniques being potentially helpful to other high-resolution studies, I suggest publication after some major revisions.

In particular, a more critical interpretation of the results and more diverse discussion is suggested. The authors should seek for going a step further compared to the already existing and increasing number of studies being conducted on that topic, i.e. a more quantitative connection to climate variability of the recent past in the region studied. In this context, I strongly suggest some combination with cave monitoring results making the story (and in particular the proxy dependencies) more reliable.

---

## Referee Comment (RC2) · Anonymous Referee #2 · 19 May 2016

[oneside,english]book mathptmx [T1]fontenc [latin1]inputenc [a4paper]geometry verbose,tmargin=1.5in,bmargin=1.5in,lmargin=1in,rmargin=1in babel setspace [authoryear]natbib [unicode=true, book-marks=true,bookmarksnumbered=false,bookmarksopen=false, break-links=false,pdfborder=0 0 0,backref=page,colorlinks=false] hyperref breakurl

**Review of doi:10.5194/cp-2016-18**

**Title:** Detection and origin of different types of annual laminae in recent stalagmites from Zoolithencave, southern Germany: Evaluation of the potential for quantitative reconstruction of past precipitation variability

**Authors:** Dana Felicitas Christine Riechelmann et al.

**Summary:** This article is based on three stalagmites from Zoolithencave in southern Germany and the potential of using these samples for high resolution (annual) reconstruction of climate. Absolute chronology is challenging but important constraints are placed on the sample-growth period. Zoo-rez-1 and Zoo-rez-2 are both very young: i) A C14 bomb signal is visible in Zoo-rez-1; ii) C-14 dated charcoal beneath Zoo-rez-2 is dated (calibrated, $1\sigma$) to between 1671 and 1951 AD. Unfortunately for U-Th chronology, the young age is combined with initial Thorium contamination. C14 measurements in speleothems (3 measurements here) are complicated by the need to know the dead carbon fraction.

However, there are important constraints on the growth period. From the charcoal C14 date it is known that the oldest stalagmite growth layers are less than 345 years old (the charcoal is located beneath the start of the visible laminae).

Multiple measurements/observations are used to show evidence of laminations in the sample (i.e. visible layers, fluorescence and trace-element measurements). Counting of visible laminae in samples 1 and 2 reveal 161 and 165 laminae respectively. The stated aims of the paper are: "i) to test the potential of different analytical methods to detect annual laminae in speleothems, ii) analyze the origin of the different types of laminae, and iii) evaluate their potential as climate proxies".

**Opinion:** In it's current form I am not in favour of the publication of this article. Below, I give some specific examples of why.

Overall, there is too much material, it is difficult to follow in a logical/critical way and much of it's content is not discussed in a sufficiently robust or well-referenced manner.

The multiple sources of laminations (visible laminae, fluorescence, elemental) are of definite interest, especially given the availability of meteorological data and a tree-ring record in the area (although the location of the tree ring record relative to the stalagmites isn't clear, a map would be useful) covering some of the ~ 170 annual speleothem laminae.

My feeling/recommendation is that a significant amount of material should be pruned/removed (fine for it to be included in a subsequent follow up publication). The smaller number of remaining points for discussion should be dealt with more robustly.

Example of points that I disagree with:

- The arguments and the article in general is often difficult to follow. A feature of too many unnecessary details at times yet missing information or missing logical argumentation for important points.

- e.g. **line 478**: "We interpret annual lamina thickness as a proxy for past precipitation. Thus we correlated the lamina thickness series to instrumental data...". i) use 'rainfall' not 'precipitation' which can be confused with mineral precipitation. ii) Why rainfall alone? What is the evidence? Why not temperature or a combination of temperature and rainfall? Choose which discussion is important and discuss it robustly.

- e.g. **line 490**: "A probable reason for the three stalagmites to stop growing in 1970 could be that further exploration of the deeper parts of the cave started in 1971. Furthermore, the correlation between precipitation of all individual months....was calculated". **Why** should three stalagmites stop growing because further exploration of deeper parts of the cave started in 1971? Perhaps there is a logical explanation, but it isn't given? Justify this point if it is important or remove this statement.

[Figure]

- The above statement (no growth after 1970) contradicts the statement on line 120: "The stalagmite was fed by an active drip when it was sampled suggesting recent growth." These things need to be ironed out so that a more coherent, robust and easy to follow narrative is available to the reader.

- The argument provided for sample growth ending in 1970 is based on $^{14}C - activity$ values from the top section of Zoo-rez-1: "Since the increase in 14C in 244 Zoo-rez is large (compare e.g., Noronha et al., 2015), the peak is near the maximum of the atmospheric 14C values. Thus, we suggest that the highest 14C value corresponds to about 1967 AD and attributed a 5 years uncertainty". In my opinion this is not robust, e.g. how large does the increase in 14C need to be before we can decide that their C14 sample 1.3 is co-eval with the C14 bomb peak? 14C could still be on it's way up? This is another example of where I believe the authors need to make choices: e.g. either focus efforts and discussion on chronology/C14 or acknowledge that there is some uncertainty on chronology (not huge based on the charcoal C14 age) and, instead, focus on other aspects (e.g. trace-elements).

- **Lines 213 to 250** and subsequently **lines 292 to 347** are dedicated to chronology. This is long, doesn't achieve very much and isn't always factually correct. **Line 213 (and table 1**) both state that the corrected age of the U-Th sample closest to the top (0.1-0.6 cm dft) is $4.669 \pm 0.9998\,[ka]$. Clearly this cannot be right - either the uncertainty assessment is wrong or the other conclusions are wrong (e.g. that the sample grew sometime between 345 years ago and 1970). Instead, I would recommend, briefly, focussing on the two pertinent pieces of chronological information: i) a piece of charcoal x cms beneath the laminations in sample 2 is at most 345 years old (in agreement with the lamination counting); and (ii) that the bomb spike is present at the top of the sample. Details of the U-Th chronology should be given too but this should be brief (simply stating the issues of the young age of the sample combined with initial detrital contamination). It could

potentially also provide a 232/230 value for the detrital contamination (relative to bulk earth). Some/all of this should be possible in half the amount of the original text.

- Line 271 indicates a laminae thickness range from 64 to $256\mu m$. Figure 4 seems to show laminations that are > 1mm. Unless I'm misreading figure 4? It could be useful to annotate fig 4 to show what the authors believe is an annual lamination.

---

## Author Comment (AC1) · 15 Jun 2016

The reviewer considers the paper interesting, undoubtedly important and relevant to various climate studies highlighting the general quality of the manuscript. We agree with the reviewer that a high resolution cave monitoring would be very useful to obtain additional information about the processes affecting the formation of different lamina types as well as speleothem climate proxies. Unfortunately, such data are not available. In the revised manuscript, we will discuss the potential problems and uncertainties in our interpretation in more detail. The reviewer is also correct that the presented chronology is not very robust yet. We will improve this with five additional 14C ages in order to better detect the shape of the bomb peak. In addition, we will discuss the uncertainties of the chronology in more detail in the revised manuscript. We will also better explain the different statistical approaches, in particular the methods conventionally applied in dendrochronology, and motivate why we use these methods. The discussion will be extended and will include all potential interpretations of the formation of the laminae. The suggestion to determine more 230Th/U ages on Zoo-rez-2 is generally good. However, we know from other stalagmites from this cave containing less detrital material, that even samples older than 1000 years are difficult to date due to the very low U-content. The reviewer is also correct with his/her comment on Figure 4 that is not the best example. We will replaced this figure by another picture better showing the laminae. We agree that a direct comparison of petrographic and elemental laminae would be better. Unfortunately, speleothem Zoo-rez was already sampled and analysed prior to this study, and we had to use the existing thin sections. The elemental measurements were performed on the opposite site. Therefore, a direct comparison is not possible and made the wiggle matching approach necessary. This will be explained in more detail in the revised manuscript. Thank you for your constructive feedback.

---

## Author Comment (AC2) · 15 Jun 2016

We thank the reviewer for his/her constructive comments. We agree that the current version of the manuscript contains too much material and is generally too long. Therefore, we will substantially shorten the manuscript. For instance, since the application of both methods was not successful, the discussion of 230Th/U- dating will be much shorter in the revised version, and the UV-scanning part will be deleted completely. This will considerably streamline the manuscript. As mentioned in our reply to reviewer

1, we will also improve the discussion with a more detailed interpretation and discussion of the proxy data. Furthermore, we will improve the chronology by five additional 14C ages. Thank you for these helpful comments.

In summary, we consider the comments of both reviewers very useful and are convinced that we are able to accordingly modify the manuscript.

---

## Author Comment (AC3) · 15 Jun 2016

Yuval Burstyn considers the paper very interesting. He is correct that Orland et al. (2012) is a more updated reference, and we will add this to the revised paper. The point concerning the lamina classification will not be changed because "mineralogical" is defined as the alteration of aragonite and calcite on a seasonal scale, whereas "petrographic" refers to a change in calcite fabric on a seasonal scale.